# Instances and Labels: Hierarchy-aware Joint Supervised Contrastive Learning for Hierarchical Multi-Label Text Classification

**Simon Chi Lok U**[1*]    **Jie He**[1*]    **Víctor Gutiérrez-Basulto**[2]    **Jeff Z. Pan**[1†]

[1]University of Edinburgh    [2]Cardiff University

c.l.u@sms.ed.ac.uk, j.he@ed.ac.uk
gutierrezbasultov@cardiff.ac.uk, j.z.pan@ed.ac.uk

## Abstract

Hierarchical multi-label text classification (HMTC) aims at utilizing a label hierarchy in multi-label classification. Recent approaches to HMTC deal with the problem of imposing an overconstrained premise on the output space by using contrastive learning on generated samples in a semi-supervised manner to bring text and label embeddings closer. However, the generation of samples tends to introduce noise as it ignores the correlation between similar samples in the same batch. One solution to this issue is supervised contrastive learning, but it remains an underexplored topic in HMTC due to its complex structured labels. To overcome this challenge, we propose HJCL, a **H**ierarchy-aware **J**oint Supervised **C**ontrastive **L**earning method that bridges the gap between supervised contrastive learning and HMTC. Specifically, we employ both instance-wise and label-wise contrastive learning techniques and carefully construct batches to fulfill the contrastive learning objective. Extensive experiments on four multi-path HMTC datasets demonstrate that HJCL achieves promising results and the effectiveness of Contrastive Learning on HMTC. Code and data are available at https://github.com/simonucl/HJCL.

## 1 Introduction

Text classification is a fundamental problem in natural language processing (NLP), which aims to assign one or multiple categories to a given document based on its content. The task is essential in many NLP applications, e.g. in discourse relation recognition (Chan et al., 2023), scientific document classification (Sadat and Caragea, 2022), or e-commerce product categorization (Shen et al., 2021). In practice, documents might be tagged with multiple categories that can be organized in a concept hierarchy, such as a taxonomy of a knowledge graph (Pan et al., 2017b,a), cf. Figure 1. The

---

*The first two authors contributed equally.
† Corresponding author.

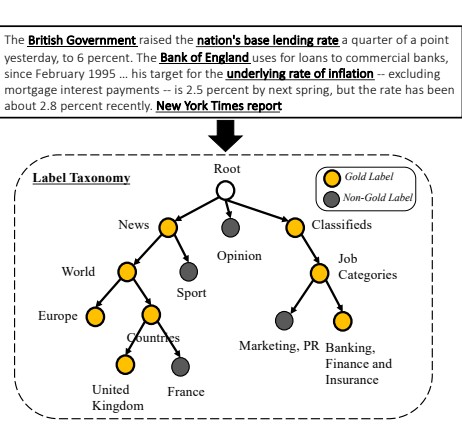

Figure 1: Example of an input sample and its annotated labels from the New York Times dataset (Sandhaus, 2008). The Label Taxonomy is a subgraph of the actual hierarchy.

task of assigning multiple hierarchically structured categories to documents is known as *hierarchical multi-label text classification* (HMTC).

A major challenge for HMTC is how to semantically relate the input sentence and the labels in the taxonomy to perform classification based on the hierarchy. Recent approaches to HMTC handle the hierarchy in a global way by using graph neural networks to incorporate the hierarchical information into the input text to pull together related input embeddings and label embeddings in the same latent space (Zhou et al., 2020; Deng et al., 2021; Chen et al., 2021; Wang et al., 2022b; Jiang et al., 2022). At the inference stage, most global methods reduce the learned representation into level-wise embeddings and perform prediction in a top-down fashion to retain hierarchical consistency. However, these methods ignore the correlation between labels at different paths (with varying lengths) and different levels of abstraction.

To overcome these challenges, we develop a method based on contrastive learning (CL) (Chen et al., 2020). So far, the application of contrastive learning in hierarchical multi-label classification

has received very little attention. This is because it is difficult to create meaningful positive and negative pairs: given the dependency of labels on the hierarchical structure, each sample could be characterized with multiple labels, which makes it hard to find samples with the exact same labels (Zheng et al., 2021). Previous endeavors in text classification with hierarchically structured labels employ data augmentation methods to construct positive pairs (Wang et al., 2022a; Long and Webber, 2022). However, these approaches primarily focus on pushing apart inter-class labels within the same sample but do not fully utilize the intra-class labels across samples. A notable exception is the work by Zhang et al. (2022a) in which CL is performed across hierarchical samples, leading to considerable performance improvements. However, this method is restricted by the assumption of a fixed depth in the hierarchy i.e., it assumes all paths in the hierarchy have the same length.

To tackle the above challenges, we introduce a supervised contrastive learning method, **HJCL**, based on utilising in-batch sample information for establishing the label correlations between samples while retaining the hierarchical structure. Technically, HJCL aims at achieving two main goals: 1) For instance pairs, the representations of intra-class should obtain higher similarity scores than inter-class pairs, meanwhile intra-class pairs at deeper levels obtain more weight than pairs at higher levels. 2) For label pairs, their representations should be pulled close if their original samples are similar. This requires careful choices between positive and negative samples to adjust the contrastive learning based on the hierarchical structure and label similarity. To achieve these goals, we first adopt a text encoder and a label encoder to map the embeddings and hierarchy labels into a shared representation space. Then, we utilize a multi-head mechanism to capture different aspects of the semantics to label information and acquire label-specific embeddings. Finally, we introduce two contrastive learning objectives that operate at the instance level and the label level. These two losses allow HJCL to learn good semantic representations by fully exploiting information from in-batch instances and labels. We note that the proposed contrastive learning objectives are aligned with two key properties related to CL: uniformity and alignment (Wang and Isola, 2020). Uniformity favors feature distribution that preserves maximal mutual information between the

representations and task output, i.e., the hierarchical relation between labels. Alignment refers to the encoder being able to assign similar features to closely related samples/labels. We also emphasize that unlike previous methods (Zhang et al., 2022a), our approach has no assumption on the depth of the hierarchy.

Our main contributions are as follows:

- We propose HJCL, a representation learning approach that bridges the gap between supervised contrastive learning and Hierarchical Multi-label Text Classification.
- We propose a novel supervised contrastive loss on hierarchical structure labels that weigh based on both hierarchy and sample similarity, which resolves the difficulty of applying vanilla contrastive in HMTC and fully utilizes the label information between samples.
- We evaluate HJCL on four multi-path datasets. Experimental results show its effectiveness. We also carry out extensive ablation studies.

## 2 Related Work

**Hierarchical Multi-label Text Classification** Existing HMTC methods can be divided into two groups based on how they utilize the label hierarchy: local or global approaches. The local approach (Kowsari et al., 2017; Banerjee et al., 2019) reuses the idea of flat multi-label classification tasks and trains unique models for each level of the hierarchy. In contrast, global methods treat the hierarchy as a whole and train a single model for classification. The main objective is to exploit the semantic relationship between the input and the hierarchical labels. Existing methods commonly use reinforcement learning (Mao et al., 2019), meta-learning (Wu et al., 2019), attention mechanisms (Zhou et al., 2020), information maximization (Deng et al., 2021), and matching networks (Chen et al., 2021). However, these methods learn the input text and label representations separately. Recent works have chosen to incorporate stronger graph encoders (Wang et al., 2022a), modify the hierarchy into different representations, e.g. text sequences (Yu et al., 2022), or directly incorporate the hierarchy into the text encoder (Jiang et al., 2022; Wang et al., 2022b). To the best of our knowledge, HJCL is the first work to utilize supervised contrastive learning for the HMTC task.

**Contrastive Learning** In HMTC, there are two major constraints that make challenging for *su-*

*pervised contrastive learning* (SCL) (Gunel et al., 2020) to be effective: multi-label and hierarchical labels. Indeed, SCL was originally proposed for samples with single labels, and determining positive and negative sets becomes difficult. Previous methods resolved this issue mainly by reweighting the contrastive loss based on the similarity to positive and negative samples (Suresh and Ong, 2021; Zheng et al., 2021). Note that the presence of a hierarchy exacerbates this problem. ContrastiveIDRR (Long and Webber, 2022) performed semi-supervised contrastive learning on hierarchy-structured labels by contrasting the set of all other samples from pairs generated via data augmentation. Su et al. (2022b) addressed the sampling issue using a $k$NN strategy on the trained samples. In contrast to previous methods, HJCL makes further progress by directly performing supervised contrastive learning on in-batch samples. In a recent study in computer vision, HiMulConE (Zhang et al., 2022a) proposed a method similar to ours that focused on hierarchical multi-label classification with a hierarchy of fix depth. However, HJCL does not impose constraints on the depth of the hierarchy and achieves this by utilizing a multi-headed attention mechanism.

## 3 Background

**Task Formulation**    Let $\mathcal{Y} = \{y_1, \ldots, y_n\}$ be a set of labels. A *hierarchy* $\mathcal{H} = (T, \tau)$ is a labelled tree with $T = (V, E)$ a tree and $\tau : V \to \mathcal{Y}$ a labelling function. For simplicity, we will not distinguish between the node and its label, i.e. a label $y_i$ will also denote the corresponding node. Given an input text $\mathcal{X} = \{\mathbf{x}_1, \ldots, \mathbf{x}_m\}$ and a hierarchy $\mathcal{H}$, the *hierarchical multi-label text classification (HMTC) problem* aims at categorizing the input text into a set of labels $Y \subseteq \mathcal{Y}$, i.e., at finding a function $\mathcal{F}$ such that given a hierarchy, it maps a document $\mathbf{x}_i$ to a label set $Y \subseteq \mathcal{Y}$. Note that, as shown in Figure 1, a label set $Y$ could contain elements from different paths in the hierarchy. We say that a label set $Y$ is *multi-path* if we can partition $Y$ (modulo the root) into sets $Y^1, \ldots, Y^k$, $k \geq 2$, such that each $Y^i$ is a path in $\mathcal{H}$.

**Multi-headed Attention**    Vaswani et al. (2017) extended the standard attention mechanism (Luong et al., 2015) to allow the model to jointly attend to information from different representation subspaces at different positions. Instead of computing a single attention function, this method first

projects the query $Q$, key $K$ and value $V$ onto $h$ different heads and an attention function is applied individually to these projections. The output is a linear transformation of the concatenation of all attention outputs: The multi-headed attention is defined as follows (Lee et al., 2018):

$$Multihead(Q, K, V) = W^O \left[ O_1 || O_2 || \ldots || O_h \right] \tag{1}$$

where $O_j = Attention(QW_j^q, KW_j^k, VW_j^v)$, and $W_j^q, W_j^k \in \mathbb{R}^{d_q \times d_q^h}$, $W_j^v \in \mathbb{R}^{d_v \times d_v^h}$ and $W^O \in \mathbb{R}^{hd_v^h \times d}$ are learnable parameters in the multi-head attention. $||$ represents the concatenation operation, $d_q^h = d_q/h$ and $d_v^h = d_v/h$.

**Supervised Contrastive Learning**    Given a mini-batch with $m$ samples and $n$ labels, we define the set of label embeddings as $Z = \{z_{ij} \in \mathbb{R}^d \mid i \in [1, m], j \in [1, n]\}$ and the set of ground-truth labels as $Y = \{y_{ij} \in \{0, 1\} \mid i \in [1, m], j \in [1, n]\}$. Each label embedding can be seen as an independent instance and can be associated to a label $\{(z_{ij}, y_{ij})\}_{ij}$. We further define $I = \{z_{ij} \in Z \mid y_{ij} = 1\}$ as the gold label set. Given an anchor sample $z_{ij}$ from $I$, we define its positive set as $\mathcal{P}_{ij} = \{z_{kj} \in I \mid y_{kj} = y_{ij} = 1\}$ and its negative set as $\mathcal{N}_{ij} = I \setminus \{\{z_{ij}\} \cup \mathcal{P}_{ij}\}$. The supervised contrastive learning loss (SupCon) (Khosla et al., 2020) is formulated as follows:

$$\mathcal{L}_{con} = \sum_{z_{ij} \in I} \frac{-1}{|\mathcal{P}_{ij}|} \sum_{z_p \in \mathcal{P}_{ij}} log \frac{\exp(z_{ij} \cdot z_p/\tau)}{\sum_{z_a \in \mathcal{P}_{ij} \cup \mathcal{N}_{ij}} \exp(z_{ij} \cdot z_a/\tau)} \tag{2}$$

## 4 Methodology

The overall architecture of HJCL is shown in Fig. 2. In a nutshell, HJCL first extracts a label-aware embedding for each label and the tokens from the input text in the embedding space. HJCL combines two distinct types of supervised contrastive learning to jointly leverage the hierarchical information and the label information from in-batch samples.: (i) Instance-level Contrastive Learning and (ii) Hierarchy-aware Label-enhanced Contrastive Learning (**HiLeCon**).

### 4.1 Label-Aware Embedding

In the context of HMTC, a major challenge is that different parts of the text could contain information related to different paths in the hierarchy. To overcome this problem, we first design and extract label-aware embeddings from input texts, with the

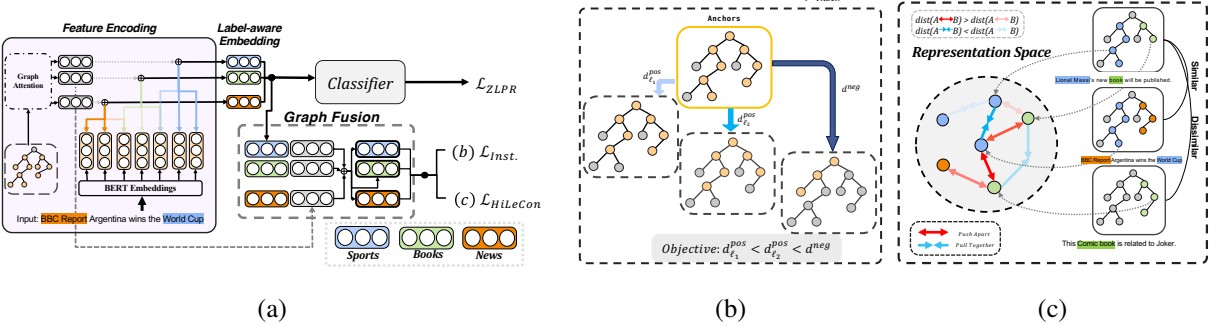

Figure 2: The model architecture for HJCL. The model is split into three parts: (a) shows the multi-headed attention and the extraction of label-aware embeddings; parts (b) and (c) show the instance-wise and label-wise contrastive learning. The legend on the lower left of (a) shows the labels corresponding to each color. We use different colors to identify the strength of contrast: the lighter the color, the less pushing/pulling between two instances/labels.

objective of learning the unique label embeddings between labels and sentences in the input text.

Following previous work (Wang et al., 2022a; Jiang et al., 2022), we use BERT (Devlin et al., 2019) as text encoder, which maps the input tokens into the embedding space: $H = \{h_1, \ldots, h_m\}$, where $h_i$ is the hidden representation for each input token $x_i$ and $H \in \mathbb{R}^{m \times d}$. For the label embeddings, we initialise them with the average of the BERT-embedding of their text description, $Y' = \{\mathbf{y}'_1, \ldots, \mathbf{y}'_n\}, Y' \in \mathbb{R}^{n \times d}$. To learn the hierarchical information, a graph attention network (GAT) (Velickovic et al., 2018) is used to propagate the hierarchical information between nodes in $Y'$.

After mapping them into the same representation space, we perform multi-head attention as defined at Eq. 1, by setting the $i^{th}$ label embedding $y_i$ as the query $Q$, and the input tokens representation $H$ as both the key and value. The label-aware embedding $g_i$ is defined as follows: $\mathbf{g}_i = Multihead(\mathbf{y}'_i, H, H)$, where $i \in [1, n]$ and $\mathbf{g}_i \in \mathbb{R}^d$. Each $\mathbf{g}_i$ is computed by the attention weight between the label $y_i$ and each input token in $H$, then multiplied by the input tokens in $H$ to get the label-aware embeddings. The label-aware embedding $\mathbf{g}_i$ can be seen as the pooled representation of the input tokens in $H$ weighted by its semantic relatedness to the label $y_i$.

### 4.2 Integrating with Contrastive Learning

Following the general paradigm for contrastive learning (Khosla et al., 2020; Wang et al., 2022a), the learned embedding $g_i$ has to be projected into a new subspace, in which contrastive learning takes place. Taking inspiration from Wang et al. (2018) and Liu et al. (2022), we fuse the label representa-

tions and the learned embeddings to strengthen the label information in the embeddings used by contrastive learning, $a_i = [\mathbf{g}_i || \mathbf{y}'_i] \in \mathbb{R}^{2d}$. An attention mechanism is then applied to the final representation $z_i = \alpha_i^T H$, where $z_i \in \mathbb{R}^d, \alpha_i \in \mathbb{R}^{m \times 1}$, $\alpha_i = softmax(H(\mathbf{W}_a a_i + \mathbf{b}_a))$ ($\mathbf{W}_a \in \mathbb{R}^{d \times 2d}$ and $\mathbf{b}_a \in \mathbb{R}^d$ are trainable parameters)

**Instance-level Contrastive Learning** For instance-wise contrastive learning, the objective is simple: the anchor instances should be closer to the instances with similar label-structure than to the instances with unrelated labels, cf. Fig. 2. Moreover, the anchor nodes should be closer to positive instance pairs at deeper levels in the hierarchy than to positive instance pairs at higher levels. Following this objective, we define a distance inequality: $dist^{pos}_{\ell_1} < dist^{pos}_{\ell_2} < dist^{neg}$, where $1 \leq \ell_2 < \ell_1 \leq L$ and $dist^{pos}_\ell$ is the distance between the anchor instance $X_i$ and $X_\ell$, which have the same labels at level $\ell$.

Given a mini-batch for instances $\{(Z_i, Y_i)\}_n$, where $Z_i = \{z_{ij} \mid j \in [0, n]\}, Z_i \in \mathbb{R}^{n \times d}$ contains the label-aware embeddings for sample $i$, we define their subsets at level $\ell$ as $Z^\ell_i = \{z_{ij} \mid z_{ij} \in Z_i, depth(y_{ij}) \leq \ell\}, Y^\ell_i = \{y_{ij} \mid depth(y_{ij}) \leq \ell\}$.

$$\mathcal{L}^{level}(Z^\ell_i, Z^\ell_j) = \log \frac{exp(X^\ell_i \cdot X^\ell_j / \tau)}{\sum_{Z_k \in \mathcal{N}_\ell \backslash i} exp(X^\ell_i \cdot X^\ell_k / \tau)}$$

where $X^\ell_i = average(Z^\ell_i)$ and $X^\ell_i \in \mathbb{R}^d$ is the mean pooling representation of $Z^\ell_i \in \mathbb{R}^{n^\ell \times d}$.

$$\mathcal{L}_{\text{Inst.}} = \frac{1}{L} \sum_l^L \frac{-1}{|\mathcal{P}_\ell|} \sum_{i \in I} \sum_{Z_j \in \mathcal{P}_\ell} \mathcal{L}^{level}(Z^\ell_i, Z^\ell_j) \cdot exp(\frac{1}{|L| - \ell})$$

where $L = \{1, \ldots, \ell_h\}$ is the set of levels in the taxonomy, $|L|$ is the maximum depth and the term

$exp(\frac{1}{|L|-\ell})$ is a penalty applied to pairs constructed from deeper levels in the hierarchy, forcing them to be closer than pairs constructed from shallow levels.

**Label-level Contrastive Learning**   We will also introduce label-wise contrastive learning. This is possible due to our extraction of label-aware embeddings in Section 4.1, which allows us to learn each label embedding independently. Although Equation 2 performs well in multi-class classification (Zhang et al., 2022b), it is not the case for multi-label classification with hierarchy. (1) It ignores the semantic relation from their original sample $\{\mathcal{X}_i, \mathcal{X}_k\}$. (2) $\mathcal{N}_{ij}$ contains the label embeddings from the same samples but with different classes, $z_{ik}$. Pushing apart labels that are connected in the hierarchy could damage the classification performance. To bridge this gap, we propose a **Hi**erarchy-aware **L**abel-**E**nhanced **Con**trastive Loss Function (**HiLeCon**), which carefully weighs the contrastive strength based on the relatedness of the positive and negative labels with the anchor labels. The basic idea is to weigh the degree of contrast between two label embeddings $z_i$, $z_j$ by their samples' label similarity, $Y_i, Y_j \in \{0, 1\}^n$. In particular, in supervised contrastive learning, the gold labels for the samples from where the label pairs come can be used for their similarity measurement. We will use a variant of the Hamming metric that treats differently labels occurring at different levels of the hierarchy, such that pairs of labels at higher level should have a larger semantic difference than pairs of labels at deeper levels. Our metric between $Y_i$ and $Y_j$ is defined as follows:

$$\rho(Y_i, Y_j) = \sum_{k=0}^{n} dist(y_{ik}, y_{jk})$$

$$dist(y_{ik}, y_{jk}) = \begin{cases} |L| - \ell_k + 1 & y_{ik} \neq y_{jk} \\ 0 & \text{Otherwise} \end{cases}$$

where $\ell_i$ is the level of the $i$-th label in the hierarchy. For example, the distance between *News* and *Classifields* in Figure 1 is 4, while the distance between *United Kingdom* and *France* is only 1. Intuitively, this is the case because *United Kingdom* and *France* are both under *Countries*, and samples with these two labels could still share similar contexts relating to the *World News*.

We can now use our metric to set the weight between positive pairs $z_{ij} \in \mathcal{P}_{ij}$ and negative pairs $z_{ik} \in \mathcal{N}_{ij}$ in Eq. 2:

$$\sigma_{ij} = 1 - \frac{\rho(Y_i, Y_j)}{C}, \gamma_{ik} = \rho(Y_i, Y_k) \quad (3)$$

where $C = \rho(\mathbf{0}_n, \mathbf{1}_n)$[1] is used to normalize the $\sigma_{ij}$ values. HiLeCon is then defined as

$$\mathcal{L}_{\text{HiLeCon}} = \frac{1}{N} \sum_{z_{ij} \in I} \frac{-1}{|\mathcal{P}_{ij}|} \sum_{z_p \in \mathcal{P}_{ij}} \quad (4)$$

$$[\log \frac{\sigma_{ij} f(z_{ij}, z_p)}{\sum_{z_a \in \mathcal{P}_{ij}} \sigma_{ij} f(z_{ij}, z_a) + \sum_{z_k \in \mathcal{N}_{ij}} \gamma_{ik} f(z_{ij}, z_k)}]$$

where $n$ is the number of labels and $f(\cdot, \cdot)$ is the exponential cosine similarity measure between two embeddings. Intuitively, in $\mathcal{L}_{\text{HiLeCon}}$ the label embeddings with similar gold label sets should be close to each other in the latent space, and the magnitude of the similarity is determined based on how similar their gold labels are. Conversely, for dissimilar labels.

## 4.3   Classification and Objective Function

At the inference stage, we flatten the label-aware embeddings and pass them through a linear layer to get the logits $s_i$ for label $i$:

$$S = (W_s([\mathbf{g}_1||\mathbf{g}_2||\dots||\mathbf{g}_n]) + b_s) \quad (5)$$

where $W_s \in \mathbb{R}^{n \times nd}, b_s \in \mathbb{R}^n, S \in \mathbb{R}^{n \times 1}$ and $S = \{s_1, \dots, s_n\}$. Instead of Binary Cross-entropy, we use the novel loss function "Zero-bounded Log-sum-exp & Pairwise Rank-based" (ZLPR) (Su et al., 2022a), which captures label correlation in multi-label classification:

$$\mathcal{L}_{\text{ZLPR}} = \log\left(1 + \sum_{i \in \Omega_{pos}} e^{-s_i}\right) + \log\left(1 + \sum_{j \in \Omega_{neg}} e^{s_j}\right)$$

where $s_i, s_j \in \mathbb{R}$ are the logits output from the Equation (5). The final prediction is as follows:

$$\mathbf{y}_{pred} = \{y_i | s_i > 0\} \quad (6)$$

Finally, we define our overall training loss function:

$$\mathcal{L} = \mathcal{L}_{\text{ZLPR}} + \lambda_1 \cdot \mathcal{L}_{\text{Inst.}} + \lambda_2 \cdot \mathcal{L}_{\text{HiLeCon}} \quad (7)$$

where $\lambda_1$ and $\lambda_2$ are the weighting factors for the Instance-wise Contrastive loss and HiLeCon.

---

[1]The maximum value for $\rho$, i.e. the distance between the empty label sets and label sets with all labels.

| Model | BGC | | AAPD | | RCV1-V2 | | NYT | |
|---|---|---|---|---|---|---|---|---|
| | Micro-F1 | Macro-F1 | Micro-F1 | Macro-F1 | Micro-F1 | Macro-F1 | Micro-F1 | Macro-F1 |
| **Hierarchy-Aware Models** | | | | | | | | |
| TextRCNN | - | - | - | - | 81.57 | 59.25 | 70.83 | 56.18 |
| HiAGM | 77.22 | 57.91 | - | - | 83.96 | 63.35 | 74.97 | 60.83 |
| HTCInfoMax | 76.84 | 58.01 | 79.64 | 54.48 | 83.51 | 62.71 | 74.84 | 59.47 |
| HiMatch | 76.57 | 58.34 | 80.74 | 56.16 | 84.73 | 64.11 | 74.65 | 58.26 |
| **Instruction-Tuned Language Model** | | | | | | | | |
| ChatGPT | 57.17 | 35.63 | 45.82 | 27.98 | $51.35_{\pm 0.18}$ | $32.20_{\pm 0.30}$ | - | - |
| **Pretrained Language Models** | | | | | | | | |
| BERT | 78.84 | 61.19 | 80.88 | 57.17 | 85.65 | 67.02 | 78.24 | 65.62 |
| HiAGM (BERT) | 79.48 | 62.84 | 80.68 | 59.47 | 85.58 | 67.93 | 78.64 | 66.76 |
| HTCInfoMax (BERT) | 79.16 | 62.94 | 80.76 | 59.46 | 85.83 | 67.09 | 78.75 | 67.31 |
| HiMatch (BERT) | 78.89 | 63.19 | 80.42 | 59.23 | 86.33 | 68.66 | - | - |
| Seq2Tree (T5) | 79.72 | 63.96 | 80.55 | 59.58 | 86.88 | 70.01 | - | - |
| HiMulConE (BERT)$^\triangle$ | 79.19 | 60.85 | 80.98 | 57.75 | 85.89 | 66.65 | 77.53 | 61.08 |
| HGCLR (BERT)$^\triangle$ | 79.22 | 64.04 | 80.95 | 59.34 | 86.49 | 68.31 | 78.86 | 67.96 |
| **HJCL** (BERT) | $\mathbf{81.30}^{\uparrow 1.58}_{\pm 0.29}$ | $\mathbf{66.77}^{\uparrow 2.73}_{\pm 0.37}$ | $\mathbf{81.91}^{\uparrow 0.96}_{\pm 0.18}$ | $\mathbf{61.59}^{\uparrow 2.01}_{\pm 0.23}$ | $\mathbf{87.04}^{\uparrow 0.16}_{\pm 0.24}$ | $\mathbf{70.49}^{\uparrow 0.48}_{\pm 0.32}$ | $\mathbf{80.52}^{\uparrow 1.66}_{\pm 0.28}$ | $\mathbf{70.02}^{\uparrow 2.06}_{\pm 0.31}$ |

Table 1: Experimental results on the four HMTC datasets. The best results are in bold and the second-best is underlined. We report the mean results across 5 runs with random seeds. Models with $^\triangle$ are those using contrastive learning. For HiAGM, HTCInfoMax and HiMatch, their works used TextRCNN (Zhou et al., 2020) as encoder in their paper, we replicate the results by replacing it with BERT. The $\uparrow$ represents the improvement to the second best model; the $\pm$ represent the *std.* between experiments.

## 5 Experiments

**Datasets and Evaluation Metrics**  We conduct experiments on four widely-used HMTC benchmark datasets, all of them consisting of multi-path labels: Blurb Genre Collection (BGC)[2], Arxiv Academic Papers Dataset (AAPD) (Yang et al., 2018), NY-Times (NYT) (Shimura et al., 2018), and RCV1-V2 (Lewis et al.). Details for each dataset are shown in Table 5. We adopt the data processing method introduced in Chen et al. (2021) to remove stopwords and use the same evaluation metrics: Macro-F1 and Micro-F1.

**Baselines**  We compare HJCL with a variety of strong hierarchical text classification baselines, such as HiAGM (Zhou et al., 2020), HTCInfoMax (Deng et al., 2021), HiMatch (Chen et al., 2021), Seq2Tree (Raffel et al., 2019), HGCLR (Wang et al., 2022a). Specifically, HiMulConE (Zhang et al., 2022a) also uses contrastive learning on the hierarchical graph. More details about their implementation are listed in A.2. Given the recent advancement in Large Language Models (LLMs), we also consider ChatGPT `gpt-turbo-3.5` (Brown et al., 2020) with zero-shot prompting as a baseline. The prompts and examples of answers from ChatGPT can be found in Appendix C.

---

[2] https://www.inf.uni-hamburg.de/en/inst/ab/lt/resources/data/blurb-genre-collection.html

## 5.1 Main Results

Table 1 presents the results on hierarchical multi-label text classification. More details can be found in Appendix A. From Table 1, one can observe that HJCL significantly outperforms the baselines. This shows the effectiveness of incorporating supervised contrastive learning into the semantic and hierarchical information. Note that although HG-CLR (Wang et al., 2022a) introduces a stronger graph encoder and perform contrastive learning on generated samples, it inevitably introduces noise into these samples and overlooks the label correlation between them. In contrast, HJCL uses a simpler graph network (GAT) and performs contrastive learning on in-batch samples only, yielding significant improvements of 2.73% and 2.06% on Macro-F1 in BGC and NYT. Despite Seq2Tree's use of a more powerful encoder, T5, HJCL still shows promising improvements of 2.01% and 0.48% on Macro-F1 in AAPD and RCV1-V2, respectively. This demonstrates the use of contrastive model better exploits the power of BERT encoder. HiMul-ConE shows a drop in the Macro-F1 scores even when compared to the BERT baseline, especially on NYT, which has the most complex hierarchical structure. This demonstrates that our approach to extracting label-aware embedding is an important step for contrastive learning for HMTC. For the instruction-tuned model, ChatGPT performs poorly, particularly suffering from minority class

performance. This shows that it remains challenging for LLMs to handle complex hierarchical information, and that representation learning is still necessary.

## 5.2 Ablation Study

| Ablation Models | RCV1-V2 | | NYT | |
|---|---|---|---|---|
| | Micro-F1 | Macro-F1 | Micro-F1 | Macro-F1 |
| Ours | **87.04** | **70.49** | **80.52** | **70.02** |
| *r.m.* Label con. | 86.67 | 69.26 | 79.90 | 69.15 |
| *r.m.* Instance con. | 86.83 | 68.38 | 79.71 | 69.28 |
| *r.m.* Both con. | 86.39 | 68.06 | 79.23 | 68.21 |
| *r.p.* BCE Loss | 86.42 | 69.24 | 79.74 | 69.26 |
| *r.m.* Graph Fusion | 85.15 | 67.61 | 79.03 | 67.25 |

Table 2: Ablation study when removing components on the RCV1-V2 and NYT datasets. *r.m.* stands for removing the component; *r.p.* stands for replace with.

To better understand the impact of the different components of HJCL on performance, we conducted an ablation study on both the RCV1-V2 and NYT datasets. The RCV1-V2 dataset has a substantial testing set, which helps to minimise experimental noise. In contrast, the NYT dataset has the largest depth. One can observe in Table 2 that without label contrastive the Macro-F1 drops notably in both datasets, 1.23% and 0.87%. The removal of HiLeCon reduces the potential for label clustering and majorly affects the minority labels. Conversely, Micro-F1 is primarily affected by the omission of the sample contrast, which prevents the model from considering the global hierarchy and learning label features from training instances of other classes, based on their hierarchical interdependencies. When both loss functions are removed, the performance declines drastically. This demonstrates the effectiveness of our dual loss approaches.

Additionally, replacing the ZMLR loss with BCE loss results in a slight performance drop, showcasing the importance of considering label correlation during the prediction stage. Further analysis between BCE loss and ZLPR is shown in Appendix B.3. Finally, as shown in the last row in Table 2, the removal of the graph label fusion has a significant impact on the performance. The projection is shown to affect the generalization power without the projection head in CL (Gupta et al., 2022). Ablation results on other datasets can be found in Appendix B.1.

## 5.3 Effects of the Coefficients $\lambda_1$ and $\lambda_2$

As shown in Equation (7), the coefficients $\lambda_1$ and $\lambda_2$ control the importance of the instance-wise and

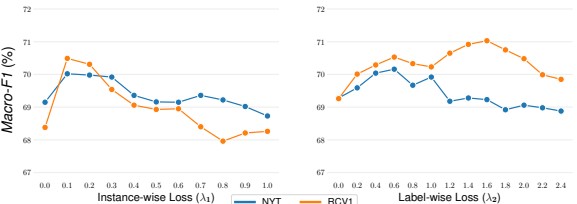

Figure 3: Effects of $\lambda_1$ (left) and $\lambda_2$ (right) on NYT and RCV1. The step size for $\lambda_1$ is 0.1 and $\lambda_2$ is 0.2. $\lambda_1$ has a smaller step size since it is more sensitive to changes.

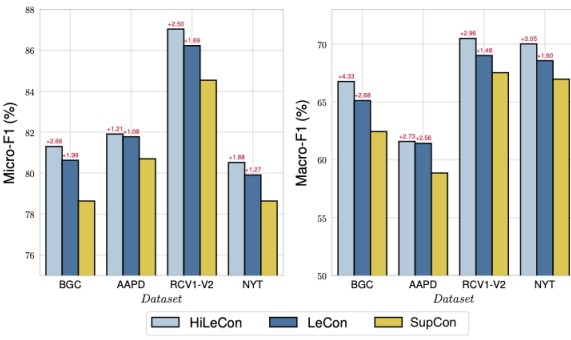

Figure 4: F1 scores on 4 different datasets with different contrastive methods. The texts above the bar show the offset between the models to the SupCon model.

label-wise contrastive loss, respectively. Figure 3 illustrates the changes on Macro-F1 when varying the values of $\lambda_1$ and $\lambda_2$. The left part of Figure 3 shows that the performance peaks with small $\lambda_1$ values and drops rapidly as these values continue to increase. Intuitively, assigning too much weight to the instance-level CL pushes apart similar samples that have slightly different label sets, preventing the models from fully utilizing samples that share similar topics. For $\lambda_2$, the F1 score peaks at 0.6 and 1.6 for NYT and RCV1-V2, respectively. We attribute this difference to the complexity of the NYT hierarchy, which is deeper. Even with the assistance of the hierarchy-aware weighted function (Eq. 3), increasing $\lambda_2$ excessively may result in overwhelmingly high semantic similarities among label embeddings (Gao et al., 2019). The remaining results are provided in Appendix B.1.

## 5.4 Effect of the Hierarchy-Aware Label Contrastive loss

To further evaluate the effectiveness of HiLeCon in Eq. 4, we conduct experiments by replacing it with the traditional SupCon (Khosla et al., 2020) or dropping the hierarchy difference by replacing the $\rho(\cdot, \cdot)$ at Eq. 3 with Hamming distance, LeCon. Figure 4 presents the obtained results and detailed

| Method | **Micro-F1** | | | | | | **Macro-F1** | | | | |
|---|---|---|---|---|---|---|---|---|---|---|---|
| | BGC | AAPD | RCV1-V2 | NYT | *p-value* | \|\| | BGC | AAPD | RCV1-V2 | NYT | *p-value* |
| HiLeCon | **81.30** | **81.91** | **87.04** | **80.52** | - | \|\| | **66.77** | **61.59** | **70.49** | **70.02** | - |
| LeCon | 80.63 | 81.78 | 86.23 | 79.91 | 3.3e-2 | \|\| | 65.12 | 61.42 | 69.01 | 68.57 | 4.0e-2 |
| SupCon | 78.64 | 80.70 | 84.54 | 78.64 | 8.3e-3 | \|\| | 62.44 | 58.86 | 67.53 | 66.97 | 2.8e-3 |

Table 3: Comparison results on different Contrastive Learning approaches on the Label embedding, performed on the 4 datasets. HiLeCon denotes our proposed method. The *p-value* is calculated by two-tailed t-tests.

| Dataset | Model | $\text{Acc}_P$ | $\text{Acc}_D$ |
|---|---|---|---|
| **NYT** | HJCL | **75.22** | **71.96** |
| | HJCL (w/o con) | 70.94 | 67.62 |
| | HGCLR | 71.26 | 70.47 |
| | BERT | 70.48 | 65.65 |
| **RCV1** | HJCL | **63.61** | **79.26** |
| | HJCL (w/o con) | 60.50 | 75.83 |
| | HGCLR | 62.99 | 78.62 |
| | BERT | 61.90 | 75.60 |

Table 4: Measurement for $\text{Acc}_P$ and $\text{Acc}_D$ on NYT and RCV1. The best scores are in bold and the second best is underlined. The formula and results on BGC and AAPD are shown in Appendix B.4.

results are shown in Table 3. HiLeCon outperforms the other two methods in all four datasets with a substantial threshold. Specifically, HiLeCon significantly outperforms the traditional SupCon in the four datasets by an absolute gain of 2.06% and 3.27% in Micro-F1 and Macro-F1, respectively. As shown in Table 3, the difference in both metrics is statistically significant with *p-value* 8.3e-3 and 2.8e-3 by a two-tailed t-test. Moreover, the improvement from LeCon in F1 scores by considering the hierarchy is 0.56% and 1.19% which are statistically significant (*p-value* = 0.033, 0.040). This shows the importance of considering label granularity with depth information.

### 5.5 Results on Multi-Path Consistency

One of the key challenges in hierarchical multi-label classification is that the input texts could be categorized into more than one path in the hierarchy. In this section, we analyze how HJCL leverages contrastive learning to improve the coverage of all meanings from the input sentence. For HMTC, the multi-path consistency can be viewed from two perspectives. First, some paths from the gold labels were missing from the prediction, meaning that the model failed to attribute the semantic information about that path from the sentences; and even if all the paths are predicted correctly, it is only able to predict the coarse-grained labels at

upper levels but missed more fine-grained labels at lower levels. To compare the performance on these problems, we measure *path accuracy* ($\text{Acc}_P$) and *depth accuracy* ($\text{Acc}_D$), which are the ratio of testing samples that have their path number and all their depth correctly predicted. (Their definitions are given in the Appendix B.4). As shown in Table 4, HJCL (and its variants) outperformed the baselines, with an offset of 2.4% on average compared with the second-best model HGCLR. Specifically, the $\text{Acc}_P$ for HJCL outperforms HGCLR with an absolute gain of 5.5% in NYT, in which the majority of samples are multi-path (cf. Table 9 in the Appendix). HJCL shows performance boosts for multi-path samples, demonstrating the effectiveness of contrastive learning.

### 5.6 Qualitative Analysis

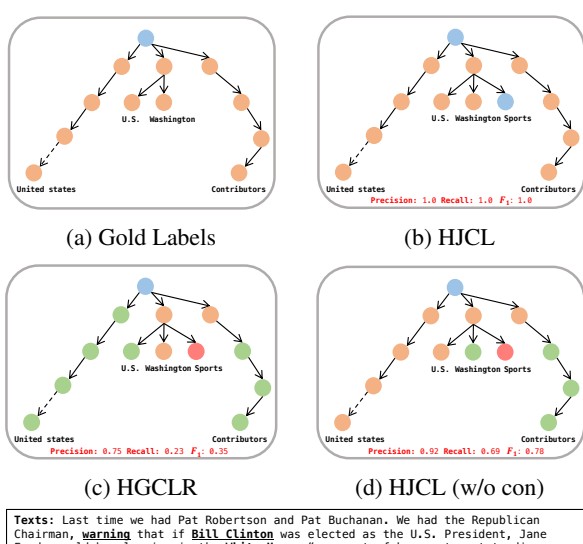

(a) Gold Labels      (b) HJCL

(c) HGCLR      (d) HJCL (w/o con)

**Texts:** Last time we had Pat Robertson and Pat Buchanan. We had the Republican Chairman, **warning** that if **Bill Clinton** was elected as the U.S. President, Jane Fonda would be sleeping in the **White House** "as guest of honor at a state dinner...

Figure 5: Case study on a sample from the NYT dataset. **Orange** represents true positive labels; **Green** represents false negative labels; **Red** represents false positive labels; **Blue** represents true negative labels. The ←-- indicates that two nodes are skipped. Part of the input texts is shown at the bottom, the full text and prediction results are in Appendix B.6.

HJCL better exploits the correlation be-

tween labels in different paths in the hierarchy with contrastive learning. For an intuition see the visualization in Figure 9, Appendix B.5. For example, the $F_1$ score of *Top/Features/Travel/Guides/Destinations/North America/United States* is only 0.3350 for the HGCLR method (Wang et al., 2022a). In contrast, our methods that fully utilised the label correlation information improved the $F_1$ score to 0.8176. Figure 5 shows a case study for the prediction results from different models. Although HGCLR is able to classify *U.S.* under *News* (the middle path), it fails to take into account label similarity information to identify the *United States* label under the *Features* path (the left path). In contrast, our models correctly identify *U.S.* and *Washington* while addressing the false positive for *Sports* under the *News* category.

## 6 Conclusion

We introduce HJCL , a combination of two novel contrastive methods that better learn the representation for embedding in Hierarchical Multi-Label Text Classification (HMTC). Our method has the following features: (1) It demonstrates that contrastive learning can help retain the hierarchy information between samples. (2) By weighting both label similarity and depth information, applying supervised contrastive learning directly at the label level shows promising improvement. (3) Evaluation on four multi-path HMTC datasets demonstrates that HJCL significantly outperforms previous baselines and shows that in-batch contrastive learning notably enhances performance. Overall, HJCL bridges the gap between supervised contrastive learning in hierarchical structured label classification tasks in general and demonstrates that better representation learning is feasible for improving HMTC performance.

In the future, we plan to look into applying our approach in some special kinds of texts, such as arguments (Saadat-Yazdi et al., 2022, 2023; Chausson et al., 2023), news (Pan et al., 2018; Liu et al., 2021; Long et al., 2020a,b) and events (Guan et al., 2023). Furthermore, we will also further develop our approach in the setting of multi-modal (Kiela et al., 2018; Chen et al., 2022b; Huang et al., 2023; Chen et al., 2022a) classification, involving both texts and images.

## Acknowledgements

This work is partially supported by the Chang Jiang Scholars Program (J2019032). We are grateful to Andrej Jovanović for helpful discussions, and to anonymous reviewers for their valuable feedback.

## Limitations

Our method is based on the extraction of a label-aware embedding for each label in the given taxonomy through multi-head attention and performs contrastive learning on the learned embeddings. Although our method shows significant improvements, the use of label-aware embeddings scales according to the number of labels in the taxonomy. Thus, our methods may not be applicable for other HMTC datasets which consist of a large number of labels. Recent studies (Ni et al., 2023) show the possible improvement of Multi-Headed Attention (MHA), which is to reduce the over-parametrization posed by the MHA. Further work should focus on reducing the number of label-aware embeddings but still retaining the comparable performance.

## Ethics Statement

An ethical consideration arises from the underlying data used for experiments. The datasets used in this paper contain news articles (e.g. NYT & RCV1-V2), books abstract (BGC) and scientific paper abstract (AAPD). These data could contain bias (Baly et al., 2018) but were not being preprocessed to address this. Biases in the data can be learned by the model, which can have significant societal impact. Explicit measures to debias data through re-annotation or restructuring the dataset for adequate representation is necessary.

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

| Dataset | $L$ | $D$ | Avg($L_i$) | Train | Dev | Test |
|---|---|---|---|---|---|---|
| BGC | 146 | 4 | 3.01 | 58,715 | 14,785 | 18,394 |
| AAPD | 61 | 2 | 4.09 | 53,840 | 1,000 | 1,000 |
| RCV1-V2 | 103 | 4 | 3.24 | 20,833 | 2,316 | 781,265 |
| NYT | 166 | 8 | 7.60 | 23,345 | 5,834 | 7,292 |

Table 5: Dataset statistics. $L$ is the number of classes. $D$ is the maximum level of hierarchy. Avg($L_i$) is the average number of classes per sample. Note that the commonly used WOS dataset (Kowsari et al., 2017) was not used as its labels are single-path only.

## A  Appendix for Experiment Settings

### A.1  Implementation Details

We implement our model using PyTorch-Lightning[3] since it is suitable for our large batches used for contrastive learning. For fair comparison, we employ the bert-base-uncased model which was used by other HMTC models to implement HJCL. The batch size is set to 80 for all datasets. Unless noted otherwise, the $\lambda_1$ and $\lambda_2$ at Eq. 7 are fixed to 0.1 and 0.5 for all datasets without any hyperparameter searching. The temperature $\tau$ is fixed at 0.1. The number of heads for multi-head attention is set to 4. We use 2 layers of GAT for hierarchy injection to BGC, AAPD and RCV1-V2; and 4 layers for NYT due to its depth. The optimizer is AdamW (Loshchilov and Hutter, 2017) with a learning rate of $3e^{-5}$. The early stopping is set to suspend training after Macro-F1 in the validation dataset and does not increase for 10 epochs. Since contrastive learning imposed stochasticity, we performed experiments with 5 random seeds. between experiments. For the baseline models, we use the hyperparameters from the original paper to replicate their results. For HiMulConE (Zhang et al., 2022a), as the model was used on the image domain, we replaced its ResNet-50 feature encoder with BERT and replicated its experiment by first training the encoder with the proposed loss and the classifier with BCE loss, with $5e^{-5}$ learning rate.

### A.2  Baseline Models

To show the effectiveness of our proposed method, HJCL, we compared it with previous HMTC works. In this section, we mainly describe baselines in recent work with strong performance.

- **HiAGM** (Zhou et al., 2020) proposes

---
[3]https://github.com/Lightning-AI/lightning

| Ablation Models | AAPD | | BGC | |
|---|---|---|---|---|
| | Micro-F1 | Macro-F1 | Micro-F1 | Macro-F1 |
| Ours | **81.91** | **61.59** | **81.30** | **66.77** |
| *r.m.* Label con. | 80.86 | 59.06 | 80.72 | 65.68 |
| *r.m.* Instance con. | 81.79 | 60.47 | 80.85 | 65.89 |
| *r.m.* Both con. | 80.47 | 58.88 | 80.57 | 65.12 |
| *r.p.* BCE Loss | 80.95 | 59.73 | 80.48 | 65.87 |
| *r.m.* Graph Fusion | 80.42 | 58.38 | 79.53 | 64.17 |

Table 6: Ablation study when removing components of on the AAPD and BGC. *r.m.* stands for removing the component; *r.p.* stands for replace with.

hierarchy-aware attention mechanism to obtain the text-hierarchy representation.

- **HTCInfoMax** (Deng et al., 2021) utilises information maximization to model the interactions between text and hierarchy.

- **HiMatch** (Chen et al., 2021) turns the problem into a matching problem by grouping the text representation with its hierarchical label representation.

- **Seq2Tree** (Yu et al., 2022) introduces a sequence-to-tree framework and turns the problem into a sequence generation task using the T5 Model (Raffel et al., 2019).

- **HiMulConE** (Zhang et al., 2022a) is the closest to our work, also performs contrastive learning on hierarchical labels, where their hierarchy has fixed height and labels are single-path only.

- **HGCLR** (Wang et al., 2022a) incorporates the hierarchy directly into BERT and performs contrastive learning on the generated positive samples.

## B  Appendix for Evaluation Result and Analysis

### B.1  Ablation study for BGC and AAPD

The ablation results for BGC and AAPD are presented in Table 6. It is worth noting that in the case of AAPD, the removal of label contrastive loss significantly affects the Micro-F1 and Macro-F1 scores in both datasets. Conversely, when the instance contrastive loss is removed, only minor changes are observed in comparison to the other three datasets. This can be primarily attributed to the shallow hierarchy of AAPD, which consists of only two levels, resulting in smaller differences

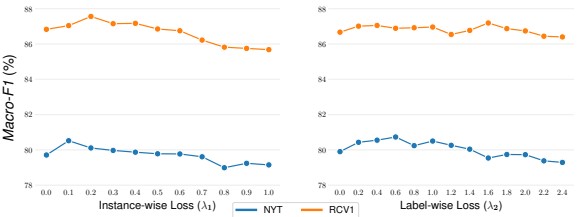

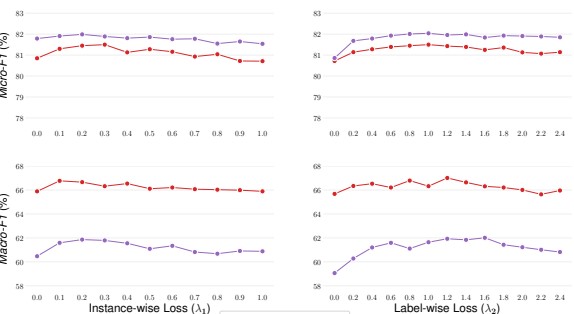

Figure 6: Effects of $\lambda_1$ (left) and $\lambda_2$ (right) on the Micro-F1 for NYT and RCV1-V2.

between instances. Furthermore, the results in Table 1 demonstrate that the substantial improvement in Macro-F1 for AAPD can be attributed to HiLe-Con, further highlighting the effectiveness of our Hierarchy-Aware label contrastive method. On the other hand, the results for BGC follow a similar trend as RCV1-V2 where both have similar hierarchy structure (c.f. Table 5), where the removal of either loss leads to a comparable drop in performance. The findings presented in the last two rows of Table 6 are consistent with the performance observed in the ablation study for NYT and RCV1-V2, underscoring the importance of both ZMLR loss and graph label fusion.

## B.2 Appendix for Hyperparameter Analysis

The hyperparameter analysis for Micro-F1 scores for NYT and RCV1-V2 is shown in Figure 6. The results are aligned with the observations for Macro-F1. Moreover, the hyperparameter analysis for BGC and AAPD regarding $\lambda_1$ and $\lambda_2$ is presented in Figure 7. Consistent with the observations from the previous ablation study section, the instance loss has a minor influence on AAPD, with performance peaking at $\lambda_1 = 0.2$ and subsequently dropping. Conversely, for any value of $\lambda_2$, the performance outperforms the baseline at $\lambda_2 = 0$, highlighting its effectiveness in shallow hierarchy labels. Additionally, the changes in BGC is consistent with those observed in RCV1-V2, as depicted in Figure 3.

Figure 7: Effects of $\lambda_1$ (left) and $\lambda_2$ (right) on both Micro- and Macro-F1 scores among the testing set for BGC and AAPD.

| Loss | BCE | | ZLPR | |
|------|------|------|------|------|
| Model | Micro-F1 | Macro-F1 | Micro-F1 | Macro-F1 |
| BERT | 85.65 | 67.02 | 86.05 | 67.42 |
| HiMatch | 86.33 | 68.66 | 86.47 | 68.98 |
| HGCLR | **86.49** | 68.31 | 86.76 | 68.34 |
| HJCL | 86.42 | **69.24** | **87.04** | **70.49** |

Table 8: Experimental results on RCV-1 dataset with traditional BCE and ZLPR loss. Best results are in **bold**.

## B.3 BCE v.s ZLPR

In this paper, we replaced the commonly-used BCE by new loss function, ZLPR (Su et al., 2022a), as it presents a more balanced loss function for the multi-label classification task, achieving this by leveraging the softmax function and considering the correlations between labels, in contrast to the Sigmoid + BCE approach proposed in the original paper (Zhang et al., 2022a). We consider this characteristic to be fundamental as it aligns with our approach of emphasizing label correlations across different paths on the hierarchy.

To provide a fairer comparison, we conducted additional experiments on strong baselines, in line with our ablation study settings. We replaced their BCE loss with the ZLPR loss on the NYT and RCV1 datasets. As shown in Tables 7 and 8, ZLPR consistently demonstrated improvements across different methods, further highlighting its effectiveness in enhancing multi-label classification. On the other side, even with the integration of the ZLPR loss function, our method continues to outperform other baseline models. This shows that it is not only the adoption of the ZLPR loss function, but the overall design that allows our model to outperform the state-of-the-art.

| Loss | BCE | | ZLPR | |
|------|------|------|------|------|
| Model | Micro-F1 | Macro-F1 | Micro-F1 | Macro-F1 |
| BERT | 78.24 | 65.62 | 78.75 | 66.24 |
| HiMatch | - | - | - | - |
| HGCLR | 78.86 | 67.96 | 79.11 | 68.37 |
| HJCL | **79.74** | **69.26** | **80.52** | **70.02** |

Table 7: Experimental results on NYT dataset with traditional BCE and ZLPR loss. Best results are in **bold**.

| Dataset \ #Path | 1 | 2 | 3 | 4 | 5 |
|---|---|---|---|---|---|
| BGC | 94.36 | 5.49 | 0.15 | - | - |
| AAPD | 57.68 | 6.79 | 35.13 | 0.36 | 0.04 |
| RCV1-V2 | 85.16 | 12.2 | 2.59 | 0.05 | - |
| NYT | 49.74 | 34.27 | 15.97 | 0.03 | - |

Table 9: Path statistics (%) among all datasets.

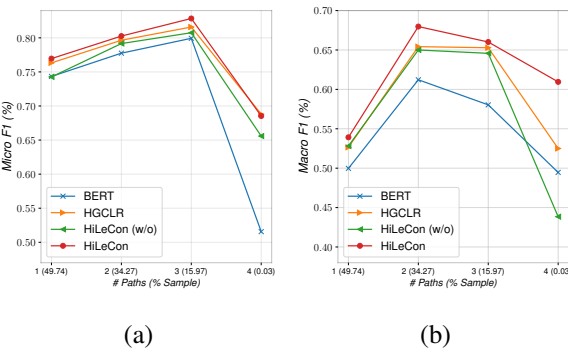

(a)                         (b)

Figure 8: (a) Micro-F1 and (b) Macro-F1 scores on testing data of NYT, grouped by paths in the hierarchy. HiLeCon is our proposed method and HiLeCon (w/o) dropped the contrastive learning function.

## B.4 Performance on Multi-Path Samples

Statistics for the number of path distributions on the four multi-path HMTC datasets are shown in Table 9. Figure 8 presents the results of the performance on samples with different paths in NYT dataset.

Before we formalize $\text{Acc}_P$ and $\text{Acc}_D$, we give the definition of some auxiliary functions. Given the testing datasets $D = \{(X_i, \hat{\mathbf{y}}_i)\}^N$ and the prediction results $\mathbf{y}_i, \forall i \leq N$, where $\hat{\mathbf{y}}_i, \mathbf{y}_i \subseteq \mathcal{Y}$, the true positive labels for each sample is defined as $\mathbf{y}_i^{Pos} = \mathbf{y}_i \cap \hat{\mathbf{y}}_i$. Then we decompose both label sets $\hat{\mathbf{y}}_i$ and $\mathbf{y}_i^{Pos}$ into disjoint sets where each set contains labels from a single path: $Path(\hat{\mathbf{y}}_i) = \{Y^i | Y^i \cap Y^j = \emptyset\}$. We say that the gold label $\hat{\mathbf{y}}_i$ and prediction $\mathbf{y}_i$ are path consistent when:

$$Path_{consistent}(\hat{\mathbf{y}}_i, \mathbf{y}_i) = \begin{cases} 1 & |Path(\hat{\mathbf{y}}_i)| = |Path(\mathbf{y}_i)| \\ 0 & \text{Otherwise} \end{cases}$$

and we say a path $Y_i$ is consistent in the predictions as:

$$Depth_{consistent}(Y^j, \mathbf{y}_i) = \begin{cases} 1 & \{Y^j \cap \mathbf{y}_i\} = Y^j \\ 0 & \text{Otherwise} \end{cases}$$

With these two definitions, we can calculate the ratio of samples and paths that are consistent with the following formulas:

$$\text{Acc}_P = \frac{\sum_{i=1}^N Path_{consistent}(\hat{\mathbf{y}}_i, \mathbf{y}_i^{Pos})}{N} \quad (8)$$

$$\text{Acc}_D = \frac{\sum_{i=1}^N \sum_{Y^j \in Path(\hat{\mathbf{y}}_i)} Depth_{consistent}(Y^j, \mathbf{y}_i^{Pos})}{\sum_{i=1}^N |Path(\hat{\mathbf{y}}_i)|} \quad (9)$$

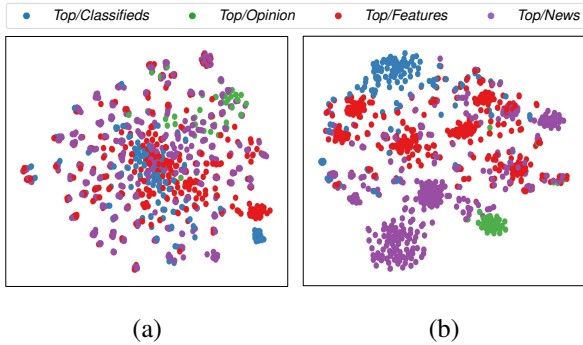

●  Top/Classifieds    ●  Top/Opinion    ●  Top/Features    ●  Top/News

(a)                         (b)

Figure 9: T-SNE visualisation (van der Maaten and Hinton, 2008) (a) HiLeCon without contrastive (b) HiLeCon. Each color represent label-aware embeddings from different path.

The $\text{Acc}_P$ is the measure for the ratio of predictions that has all the path corrected predicted; the $\text{Acc}_D$ is the measure for the ratio of paths that the prediction got it all correct. The results on multi-path consistency for BGC and AAPD are shown in Table 10.

| Dataset | Model | $\text{Acc}_P$ | $\text{Acc}_D$ |
|---|---|---|---|
| **BGC** | HJCL | **63.79** | **72.30** |
| | HJCL w/o con | 60.42 | 68.38 |
| | HGCLR | 61.46 | 70.93 |
| | BERT | 52.99 | 68.85 |
| **AAPD** | HJCL | **81.42** | **71.62** |
| | HJCL w/o con | 80.11 | 70.76 |
| | HGCLR | 80.76 | 71.59 |
| | BERT | 77.10 | 68.89 |

Table 10: Measurement for Path Accuracy and Depth Accuracy on BGC and AAPD.

## B.5 T-SNE visualisation

To qualitatively analyse the HiLeCon, we plot the T-SNE visualisation with learned label embedding across the path, as shown in Figure 9.

## B.6 Case study details

The complete news report in the NYT dataset used for the case study is shown in Figure 10. The complete set of labels for the four hierarchy plots (Figure 5) is shown in Table 11. Note that to save space, the ascendants of leaf labels are omitted since they are already self-contained within the names of the leaf labels themselves.

## C Discussion and Case Example for ChatGPT

For each prompt, the LLM is presented with input texts, label words structured in a hierarchical

Last time we had Pat Robertson and Pat Buchanan. We had the Republican Chairman, Rich Bond, **warning** that if **Bill Clinton** was elected as the **U.S. President**, Jane Fonda would be sleeping in the White House "as guest of honor at a state dinner 'for Fidel Castro. "If you liked the 1992 Republican National Convention, with "its bashing of the un-Christian, you'll love the 1996 convention. Assuming, that is, that the party wins as big as it expects in 'the upcoming **midterm elections**. Next time the powerful new committee chairmen in a Republican-controlled Senate will surely be featured. Among them are Jesse Helms, chairman of the Foreign Relations "Committee, and Alfonse D'Amato, chairman of Banking, Housing and Urban" Affairs. The Speaker of the House, Newt Gingrich, will be on the platform, **expanding on his theme that Democrats** are "the enemy of normal Americans."

-- New York Times, 1994

Figure 10: The complete input text sample used for the case study in Section 5.6.

| Gold Labels |
| --- |
| • *Top/News/U.S.* |
| • *Top/News/Washington* |
| • *Top/Features/Travel/Guides/Destinations/ North America/United States* |
| • *Top/Opinion/Opinion/Op-Ed/Contributors* |

| Models | Predictions |
| --- | --- |
| HJCL | • *Top/News/U.S.*
• *Top/News/Washington*
• *Top/Features/Travel/Guides/Destinations/ North America/United States*
• *Top/Opinion/Opinion/Op-Ed/Contributors* |
| HJCL (w/o Con) | • *Top/News/Sports*
• *Top/Features/Travel/Guides/Destinations/ North America/United States*
• *Top/Opinion/Opinion/Op-Ed* |
| HGCLR | • *Top/News/Sports*
• *Top/News/U.S.*
• *Top/Opinion* |

Table 11: Complete labels set for the case study diagram shown in Figure 5. *Orange* represent labels that are in the gold label set but some of its decedents were missing; *Red* represents the incorrect labels.

format, and a natural language command that asks it to classify the correct labels related to the texts (Wang et al., 2023). We flatten the hierarchy labels following the method used by Chan et al. (2023) in their prompt tuning approach for Discourse Relation Recognition with hierarchical structured labels. This method maintains the hierarchy dependency by connecting labels with an arrow ($\rightarrow$). For example, taking the label from BGC, the label "*World History*" appears at level-3 in the hierarchy with ascendants "*History*" and "*Nonfiction*". This label is flattened into words as "*Nonfiction $\rightarrow$ History $\rightarrow$ World History*". This dependency relation is also explicitly mentioned within the prompt. Three examples for AAPD, BGC, and RCV1-V2 are given

| Dataset | Micro-P | Micro-R | Macro-P | Macro-R | OOD |
| --- | --- | --- | --- | --- | --- |
| AAPD | 50.97 | 41.61 | 36.89 | 30.75 | 6.113 |
| BGC | 50.82 | 65.33 | 35.65 | 45.02 | 12.03 |
| RCV1 | $42.15_{\pm0.26}$ | $65.67_{\pm0.14}$ | $29.84_{\pm0.34}$ | $46.59_{\pm0.28}$ | 7.213 |
| NYT | - | - | - | - | - |

Table 12: `gpt-turbo-3.5` performance details on HMTC datasets. The Micro/Macro-P and Micro/Macro-R refers to the precision and recall for each metric respectively. The OOD refer to the ratio of "Out-of-Domain" labels in the returned answers.

in Tables 13, 14, and 15. In the experimental stage, since RCV1-V2 contains a huge testing dataset, we performed random sampling with 30,000 samples ($3 \times 10,000$) without replacement, using a random seed of 42. The performance in Table 1 for RCV1-V2 records the mean and standard deviation (std) for the three runs. As shown in Table 12, ChatGPT mainly struggles in predicting minority labels, leading to significantly lower results in Macro Precision. Meanwhile, Hallucination is a well-known problem in ChatGPT (Bang et al., 2023), and this issue also occurs in text classification, as demonstrated in the last column of Table 12, which represents the ratio of returned answers that are not within the provided categories list. Although Few-shot In-context Learning (Brown et al., 2020) may be able to mitigate this problem by providing a small subset of training samples, the flatten hierarchical labels occupy most of the tokens, and the training samples may not fit within the maximum token limit (4096 tokens). Future work on HMTC for in-context learning should focus on finding better ways to decompose and shorten the hierarchy labels.

| | |
|---|---|
| Prompt Template | `Classifiy the given text into the following categories, which`
`could belong to single or multiple categories:`
**`['Computer Science', 'Computer Science -> Performance',`**
**`'Computer Science -> Formal Languages and Automata Theory'`**
**`'Computer Science -> Robotics', ..., 'Mathematics -> Logic']`**
`Rules:`
`1. The label prediction must be consistent, which means`
`    predicting "A -> B" also needs to predict "A"`
`2. No explanation is needed, output only the categories`
`Texts:` **`[Input]`** |
| Input Texts | `In this paper we investigate the descriptional complexity of knot`
`theoretic problems and show upper bounds for planarity problem of`
`signed and unsigned knot diagrams represented by Gauss words ...`
`We study these problems in a context of automata models`
`over an infinite alphabet.` |
| Answer | `Computer Science -> Formal Languages and Automata Theory,`
`Mathematics -> Combinatorics.` |
| Gold Labels | `['cs.fl', 'cs.cc', 'cs']` |

Table 13: Example for Question and Answer from `gpt-turbo-3.5` from AAPD

| | |
|---|---|
| Prompt Template | `Classifiy the given text into the following categories, which`
`could belong to single or multiple categories:`
**`['Children's Books', 'Poetry', 'Fiction', 'Nonfiction'`**
**`'Teen & Young Adult', 'Classics', 'Humor' , ...`**
**`'Nonfiction -> History -> World History -> Asian World History']`**
`Rules:`
`1. The label prediction must be consistent, which means`
`    predicting "A -> B" also needs to predict "A"`
`2. No explanation is needed, output only the categories`
`Texts:` **`[Input]`** |
| Input Texts | `Title: Jasmine Is My Babysitter (Disney Princess).`
`Text: An original Disney Princess Little Golden Book starring`
`Jasmine as a super-fun babysitter!.... each Disney Princess`
`and shows how they relate to today's girl` |
| Answer | `['Children's Books', 'Fiction',`
`    Children's Books -> Step Into Reading']` |
| Gold Labels | `['Children's Books']` |

Table 14: Example for Question and Answer from `gpt-turbo-3.5` from BGC

| | |
|---|---|
| Prompt Template | Classifiy the given text into the following categories, which could belong to single or multiple categories: **['CORPORATE/INDUSTRIAL', 'ECONOMICS', 'GOVERNMENT/SOCIAL', 'MARKETS', 'CORPORATE/INDUSTRIAL -> STRATEGY/PLANS, ... 'CORPORATE/INDUSTRIAL -> LEGAL/JUDICIAL'** Rules: 1. The label prediction must be consistent, which means predicting "A -> B" also needs to predict "A" 2. No explanation is needed, output only the categories Texts: **[Input]** |
| Input Texts | A stand in a circus collapsed during a show in northern France on Friday, injuring about 40 people, most of them children, rescue workers said. About 25 children were injured and another 10 suffered from shock when their seating fell from under them in the big top of the Zavatta circus. One of the five adults injured was seriously hurt. |
| Answer | GOVERNMENT/SOCIAL -> DISASTERS AND ACCIDENTS |
| Gold Labels | [GOVERNMENT/SOCIAL, DISASTERS AND ACCIDENTS] |

Table 15: Example for Question and Answer from `gpt-turbo-3.5` for RCV1-V2