# OpenReview forum: "Instances and Labels: Hierarchy-aware Joint Supervised Contrastive Learning for Hierarchical Multi-Label Text Classification"
_EMNLP/2023/Conference — EMNLP 2023 Findings_

### Official Review · Reviewer_9pSu · 2023-07-27

**Typos Grammar Style And Presentation Improvements:** 1. Enhancing the formulation of the b…
**Soundness:** 4

**Excitement:**

3: Ambivalent: It has merits (e.g., it reports state-of-the-art results, the idea is nice), but there are key weaknesses (e.g., it describes incremental work), and it can significantly benefit from another round of revision. However, I won't object to accepting it if my co-reviewers champion it.

**Paper Topic And Main Contributions:**

This paper introduces Hierarchy-aware Joint Supervised Contrastive Learning (HJCL), a novel approach aimed at addressing the challenge of representation learning in Hierarchical Multi-Label Text Classification (HMTC). HJCL incorporates a ZLPR loss and two label-aware contrastive loss terms, effectively mitigating the under-exploitation of label information in traditional contrastive learning methods. Specifically, HJCL extracts label-aware embeddings from the input features through multi-head attention and devises various strategies to construct contrastive pairs based on the label hierarchy, enabling both instance-wise and label-wise contrastive learning. Extensive experiments conducted on four HMTC datasets demonstrate the superior performance of HJCL compared to other baselines, establishing its effectiveness in the recent state-of-the-art approaches.

**Questions For The Authors:**

My primary concern revolves around the extent of improvement achieved by implementing the ZLPR loss [1] in HMTC. Based on the findings from Table 2 and Table 5, when the ZLPR loss is substituted with the BCE loss, there is an approximate 1% decrease in the F1 score across all datasets. As the other baselines did not incorporate the ZLPR loss in their experiments, it would be more appropriate to consider the results obtained from the BCE loss as the main reference for a fair comparison.

[1] Su et al. ZLPR: A Novel Loss for Multi-label Classification. 2022.

**Reasons To Accept:**

1. This paper presents an effective supervised contrastive learning approach that leverages the label hierarchy to learn representations.
2. The proposed approach demonstrates outstanding performance across all benchmark datasets, surpassing the baselines by a substantial margin.
3. The experiments conducted in the study are both comprehensive and sufficient, effectively revealing the motivation behind the method.

**Reasons To Reject:**

1. Certain notations employed in this paper lack clear definitions, posing challenges to readers' comprehension.
2. The utilization of the ZLPR loss function, distinct from other baseline methods, may introduce bias and result in an unfair comparison.
3. Considering the existence of prior work focusing on hierarchical contrastive learning methods [1], the novelty of the proposed approach within the contrastive learning domain may be ambiguous.

[1] Zhang et al. Use All The Labels: A Hierarchical Multi-Label Contrastive Learning Framework. CVPR 2022.

**Reproducibility:**

4: Could mostly reproduce the results, but there may be some variation because of sample variance or minor variations in their interpretation of the protocol or method.

**Reviewer Confidence:**

3: Pretty sure, but there's a chance I missed something. Although I have a good feel for this area in general, I did not carefully check the paper's details, e.g., the math, experimental design, or novelty.

---

> ### Author Rebuttal · Authors · 2023-08-28
>
> > **Q1**: Certain notations employed in this paper lack clear definitions, posing challenges to readers' comprehension.
>
>
>
> **R1**: Thanks for the valuable comment. For the final version we will improve the presentation of the paper, focusing on consistency, self-containment, and clarity.
>
>
>
> > **Q2**: The utilization of the ZLPR loss function, distinct from other baseline methods, may introduce bias and result in an unfair comparison.
>
>
>
> **R2**: Response is put together inside **R4**.
>
>
>
> > **Q3**: Considering the existence of prior work focusing on hierarchical contrastive learning methods [1], the novelty of the proposed approach within the contrastive learning domain may be ambiguous.
>
>
>
> **R3**: Thanks for your comment on the novelty of the proposed approach. Indeed, as pointed out, HiMulConE [1] is the work most related to our submission. However, as discussed in the submission (lines 77 – 83, 184 – 191), [1] has some important shortcomings:
>
>
>
> 1. The method proposed by Zhang _et al._ [1] is restricted by the assumption of a *fixed-depth hierarchy* i.e., it assumes all paths in the hierarchy have the same length. Note that this does not apply to most HMTC datasets (e.g. NYT and BGC) as not all paths from the root to leaf have the same depth (different labels have different granularity). This limited the generalization of their approach. Our method does not have this limitation.
>
>
>
> 2. In addition to the last points, the HiMulConE method is only applied to single-path hierarchy labels (i.e., for all labels $y_{i}$ in the target label set $Y^{\text{gold}}$, there would be at most one descendant label $y_{j} \in Y^{\text{gold}}$ s.t. $depth_{y_{j}} = depth_{y_{i}} + 1$) and its contrastive method is based on this assumption. However, as shown in the statistics of the HMTC dataset (Table 7), most of the datasets are multi-path, meaning label set contain labels from more than one path. Therefore, their method is not designed for the HMTC task. Instead, our method accounts for the idea of multi-path and considers the correlation between labels at different paths.
>
>
>
> 3. To better understand our novelty, we replicated the experiment with HiMulConE, cf. Table 1. As discussed between line 454 – 460, the HiMulConE model performs even poorer than the BERT baseline. This shows that its assumption of fixed depth and considering each sample as a single embedding damages the performance.
>
>
>
> Given these identified downsides and the observed decline in performance, we strongly believe that our approach effectively bridges the contrastive learning gap in the broader context of the HMTC task. We experimentally showed this on common HMTC datasets which have multi-path structures with varying path-depths. Furthermore, the ablation study, focused on path and depth consistency (Section 5.5), further strengthened the novelty of our method compared to the state-of-the-art.
>
>
>
> > **Q4**: My primary concern revolves around the extent of improvement achieved by implementing the ZLPR loss [1] in HMTC. Based on the findings from Table 2 and Table 5, when the ZLPR loss is substituted with the BCE loss, there is an approximate 1% decrease in the F1 score across all datasets. As the other baselines did not incorporate the ZLPR loss in their experiments, it would be more appropriate to consider the results obtained from the BCE loss as the main reference for a fair comparison.
>
>
>
> **R4**: Thank you for the valuable suggestions. ZLPR presents a more balanced loss function for the multi-label classification task, achieving this by leveraging the softmax function and considering the correlations between labels, in contrast to the sigmoid + BCE approach proposed in the original paper [2]. We consider this characteristic to be fundamental as it aligns with our approach of emphasizing label correlations across different paths on the hierarchy.
>
>
>
> To provide a fairer comparison, we conducted additional experiments on strong baselines, in line with our ablation study settings. We replaced their BCE loss with the ZLPR loss on the NYT and RCV1 datasets. As shown in Tables 1 and 2, ZLPR consistently demonstrated improvements across different methods, further highlighting its effectiveness in enhancing multi-label classification. Furthermore, even with the integration of the ZLPR loss function, our method continues to outperform other baseline models. This shows that it is not only the adoption of the ZLPR loss function, but the overall design that allows our model to outperform the state-of-the-art.
>
>
>
> | Model | Micro-F1(BCE) | Macro-F1(BCE) | Micro-F1(ZLPR) | Macro-F1(ZLPR) |
> |---------|--------------|---------------|--------------|---------------|
> | BERT       | 78.24 | 65.62      | 78.75      |  66.24     |
> | HiMatch       | -  | -     | - | -     |
> | HGCLR      |  78.86 | 67.96      | 79.11  | 68.37  |
> | HJCL      |  **79.74** | **69.26**      | **80.52**      |  **70.02**     |
>
>
>
> **Table 1**: Experimental results on NYT dataset with traditional BCE and ZLPR loss. Best results are in **bold**.
>
>
>
> | Model | Micro-F1(BCE) | Macro-F1(BCE) | Micro-F1(ZLPR) | Macro-F1(ZLPR) |
> |---------|--------------|---------------|--------------|---------------|
> | BERT       | 85.65 | 67.02      | 86.05      |  67.42     |
> | HiMatch       |86.33 | 68.66     | 86.47      |  68.98     |
> | HGCLR      |  **86.49** | 68.31      | 86.76      |  68.34     |
> | HJCL      |  86.42 | **69.24**      | **87.04**      |  **70.49**     |
>
> **Table 2**: Experimental results on RCV-1 dataset with traditional BCE and ZLPR loss. Best results are in **bold**.
>
>
>
> [1] Zhang et al. Use All The Labels: A Hierarchical Multi-Label Contrastive Learning Framework. CVPR 2022.
>
> [2] Su et al. ZLPR: A Novel Loss for Multi-label Classification. 2022.

---

### Official Review · Reviewer_46re · 2023-08-04

**Soundness:** 4

**Excitement:**

3: Ambivalent: It has merits (e.g., it reports state-of-the-art results, the idea is nice), but there are key weaknesses (e.g., it describes incremental work), and it can significantly benefit from another round of revision. However, I won't object to accepting it if my co-reviewers champion it.

**Paper Topic And Main Contributions:**

This paper proposes a new method called HJCL that bridges the gap between supervised contrastive learning and HMTC. The framework also employs both instance-wise and label-wise contrastive learning techniques and carefully constructs batches to fulfill the contrastive learning objective. Extensive experiments on four multi-path HMTC datasets have shown that HJCL has achieved promising results, demonstrating the effectiveness of Contrastive Learning on HMTC.

**Reasons To Accept:**

- The paper proposes a new method called HJCL that bridges the gap between supervised contrastive learning and HMTC, which can improve the accuracy of HMTC models.
- The paper employs both instance-wise and label-wise contrastive learning techniques and carefully constructs batches to fulfill the contrastive learning objective, which can reduce the noise introduced during the generation of samples.
- The paper conducts extensive experiments on four multi-path HMTC datasets and achieves promising results, demonstrating the effectiveness of Contrastive Learning on HMTC.
- Very good visualizations and well-organized writing make the entire manuscript easy to follow.

**Reasons To Reject:**

- The authors did not release data and code to facilitate future research.


**Reproducibility:**

4: Could mostly reproduce the results, but there may be some variation because of sample variance or minor variations in their interpretation of the protocol or method.

**Reviewer Confidence:**

4: Quite sure. I tried to check the important points carefully. It's unlikely, though conceivable, that I missed something that should affect my ratings.

---

> ### Author Rebuttal · Authors · 2023-08-27
>
> > **Q1**: The authors did not release data and code to facilitate future research.
>
> **A1**: Thanks for the valuable comments. As announced in the last sentence of the abstract, we will release our code and data processing code upon acceptance. This follows the usual practice in EMNLP. As proof of this, we have uploaded our codebase with detailed documentation in an anonymized repository: https://anonymous.4open.science/r/HJCL-EMNLP.

---

### Official Review · Reviewer_Jpbo · 2023-08-05

**Soundness:** 3

**Excitement:**

2: Mediocre: This paper makes marginal contributions (vs non-contemporaneous work), so I would rather not see it in the conference.

**Missing References:**

Zhang, Tianyi, et al. "Structural Contrastive Representation Learning for Zero-shot Multi-label Text Classification." Findings of the Association for Computational Linguistics: EMNLP 2022. 2022.


Zhang, Yu, et al. "Metadata-induced contrastive learning for zero-shot multi-label text classification." Proceedings of the ACM Web Conference 2022. 2022.

Wang, Ran, and Xinyu Dai. "Contrastive learning-enhanced nearest neighbor mechanism for multi-label text classification." Proceedings of the 60th Annual Meeting of the Association for Computational Linguistics (Volume 2: Short Papers). 2022.

Huang, Wei, et al. "Exploring Label Hierarchy in a Generative Way for Hierarchical Text Classification." Proceedings of the 29th International Conference on Computational Linguistics. 2022.



**Paper Topic And Main Contributions:**

The paper presents HJCL, a Hierarchy-aware Joint Supervised Contrastive Learning method for hierarchical multi-label text classification (HMTC) that effectively addresses the issue of noisy sample generation and achieves promising results on multiple datasets, demonstrating the effectiveness of contrastive learning in HMTC.





**Questions For The Authors:**

Please consider to highlight the novelty and differences between your work and these close baselines mentioned below.

**Reasons To Accept:**

- The proposed task is interesting and has potential application in real-life scenarios.

- The paper conduct extensive experiments on multiple HMTC datasets, providing evidence of the practicality of HJCL.

- The visualization aspect is well-executed, aiding readers in gaining a better understanding of the presented findings.


**Reasons To Reject:**

The proposed method is not novel enough, as contrastive learning has been employed in various multi-label classification works (missing references). The authors should acknowledge and discuss these existing works to establish the uniqueness and importance of their approach adequately. A more comprehensive comparison with related methods is necessary to justify the novelty and significance of their proposed approach.






**Reproducibility:**

4: Could mostly reproduce the results, but there may be some variation because of sample variance or minor variations in their interpretation of the protocol or method.

**Reviewer Confidence:**

2: Willing to defend my evaluation, but it is fairly likely that I missed some details, didn't understand some central points, or can't be sure about the novelty of the work.

---

> ### Author Rebuttal · Authors · 2023-08-27
>
> > **Q1**: The proposed method is not novel enough, as contrastive learning has been employed in various multi-label classification works (missing references). The authors should acknowledge and discuss these existing works to establish the uniqueness and importance of their approach adequately. A more comprehensive comparison with related methods is necessary to justify the novelty and significance of their proposed approach.
>
> **A1**: We thank the reviewer for pointing out their concern on the novelty of the proposed approach and including related papers that utilize contrastive learning in the multi-label classification context. However, we note that our focus is specifically on **hierarchical multi-label classification (HMTC)**, which imposes an additional constraint due to the presence of labelled hierarchies. Specifically, we emphasize the issue of **multi-path labels**, and our approach aims to focus on learning the correlations between labels across different paths. Most previous works have ignored this aspect and focused solely on datasets with single-path labels [1, 4]. While preparing the submitted paper, we did come across some of the shared papers. However, we decided not to include them (given the space limitations), as we perceived their relevance to be limited. We acknowledge that contrastive learning is not a new concept in the realm of multi-label classification, but to the best of our knowledge, none of the existing literature explored the application of contrastive learning within the domain of HMTC. We did find one exception, [1], which we extensively discussed and compared with our work. In our discussion, we also elucidated the weaknesses of their approach and expounded on how our method addresses these shortcomings. This discussion can be found between lines 77 and 83, as well as lines 184 and 191.
>
> Furthermore, we introduced the concept of Label-level Contrastive Learning. In lines 335 to 344, we present a comparison between our approach and previous multi-class classification methods that integrated contrastive learning. Additional comparison between our method and these approaches is presented in the context of Table 1. The analysis of the results, articulated between lines 454 and 460, serves to bolster the novelty of our contribution.
>
> > Please consider to highlight the novelty and differences between your work and these close baselines mentioned below.
> - **C1**: Zhang, Tianyi, et al. "Structural Contrastive Representation Learning for Zero-shot Multi-label Text Classification." Findings of the Association for Computational Linguistics: EMNLP 2022.
> - **C2**: Zhang, Yu, et al. "Metadata-induced contrastive learning for zero-shot multi-label text classification." Proceedings of the ACM Web Conference 2022. 2022.
>
> **A2**: Both of these works utilize contrastive learning as the objective for zero-shot multi-label text classification, a task focused on predicting unseen labels. The first work, C1, employs randomized segmentation (RTS) to divide long texts into positive pairs. On the other hand, C2 leverages learning from web-scraped metadata. Subsequently, a standard contrastive learning function is employed for the learning process. From our understanding, these two works belong to distinct domains (zero-shot vs. hierarchical), and their primary novelty lies in their approaches to generating or sampling data not present in the training set, which is crucial for achieving zero-shot classification. Their work used the most standard CL objective, without being weighted by any similarity between the labels. This would caused problems in constructing negative batches for multi-label classification [2] as it is hard to find samples that have completely different sets of labels. This issue is also discussed in the ablation study (Section 5.4 and Table 6). Consequently, we consider it inappropriate to draw a direct comparison.
>
> - **C3**: “Wang, Ran, and Xinyu Dai. "Contrastive learning-enhanced nearest neighbor mechanism for multi-label text classification." Proceedings of the 60th Annual Meeting of the Association for Computational Linguistics (Volume 2: Short Papers). 2022.”
>
> **A3**: This work proposes a $k$nn retrieval-based method for flat multi-label classification, with the addition of a contrastive learning objective to ensure that train samples with similar labels are closer to each other. The proposed approach received comparable improvement over previous work.
>
> Firstly, even though both papers use the same dataset, it should noted that the topic of our paper and C3 are different. Our focus is on hierarchical multi-label classification, where the main challenge is how to utilize the hierarchical information in prediction; where C3 ignored the hierarchy information. This highlights a clear separation between our work and C3; that is why we did not mentioned it in the submission. Secondly, C3 proposes a contrastive learning function weighted by the dynamic coefficient $\beta_{ij}$, which is the normalization of the similarity label vectors between samples. This idea is similar to the _LeCon_ method we discussed in Section 5.4, where we analyze different types of label-wise CL and _LeCon_ is the version of _HiLeCon_ without hierarchical information. The results in Figure 4 show a consistent improvement over 4 datasets and showed statistically significance with p-value testing in Table 6.
>
> - **C4**: “Huang, Wei, et al. "Exploring Label Hierarchy in a Generative Way for Hierarchical Text Classification." Proceedings of the 29th International Conference on Computational Linguistics. 2022.”
>
> **A4**: This work proposes a generative method, PAAM-HiA-T5, for solving the HMTC problem. The proposed method involves converting the labels in a hierarchical format into a flattened one by utilizing Breadth-First Search (BFS). The T5 model is subsequently employed as a generative model, considering both text sequences and upper-level label predictions while generating labels for the current level. This approach shares similarities with the methodology discussed in reference [3], which is thoroughly compared with in our submission. However, an additional path-adaptive attention mechanism is introduced in PAAM-HiA-T5 to enhance the weighting of the currently generated path, thereby reducing path inconsistency issues. Consequently, PAAM-HiA-T5 outperforms the prior work [3], which also leveraged T5 as its foundational model.
>
> Unlike the three works above, this work does utilize the hierarchical  information and addresses the HMTC problem in a generative method, using the T5 model. It’s worth noting that our method, HJCL, outperforms the PAAM-HiA-T5 in RCV-1 and NYT (except the Micro-F1 for RCV-1 with a tiny difference) even with a weaker base model (ours: BERT-base, their: T5). The major difference is in the Macro-F1 for the NYT dataset, with an absolute 5-point higher. This highlights that the contrastive approach can better exploit base  models for the task, while enhancing the performance for minority labels.
>
> We agree with the reviewer that this reference should be included in the paper and compared with in Table 1, since  it can be seen as descendant work of Seq2Tree [3]. This will be done in the revision.
>
> [1] Zhang et al. Use All The Labels: A Hierarchical Multi-Label Contrastive Learning Framework. CVPR 2022.
>
> [2] Not All Negatives are Equal: Label-Aware Contrastive Loss for Fine-grained Text Classification (Suresh & Ong, EMNLP 2021)
>
> [3] Yu, Chao et al. “Constrained Sequence-to-Tree Generation for Hierarchical Text Classification.” Proceedings of the 45th International ACM SIGIR Conference on Research and Development in Information Retrieval (2022): n. pag.
>
> [4] Wang, Ran, and Xinyu Dai. "Contrastive learning-enhanced nearest neighbor mechanism for multi-label text classification." Proceedings of the 60th Annual Meeting of the Association for Computational Linguistics (Volume 2: Short Papers). 2022.

---

### Meta-Review · Area_Chair_ic59 · 2023-09-20

**Recommendation:** 3

**Metareview:**

Summary: This paper presents a new method called Hierarchy-aware Joint Supervised Contrastive Learning (HJCL), which aims to tackle the challenge of representation learning in Hierarchical Multi-Label Text Classification (HMTC). HJCL uses multi-head attention to extract label-aware embeddings from input features and employs different strategies to create contrastive pairs based on the label hierarchy, allowing for both instance-wise and label-wise contrastive learning. Extensive experiments conducted on four HMTC datasets demonstrate the effectiveness of the proposed approach.

Strength: This paper presents a promising method to improve the accuracy of HMTC models. The paper is well-written and easy to follow.  Extensive experiments on four multi-path HMTC datasets are conducted.  All reviewers agree that this is a solid contribution but the excitement is relatively limited.

---

### Decision · Program_Chairs · 2023-10-07

**Decision:**

Accept-Findings

**Comment:**

Summary: This paper presents a new method called Hierarchy-aware Joint Supervised Contrastive Learning (HJCL), which aims to tackle the challenge of representation learning in Hierarchical Multi-Label Text Classification (HMTC). HJCL uses multi-head attention to extract label-aware embeddings from input features and employs different strategies to create contrastive pairs based on the label hierarchy, allowing for both instance-wise and label-wise contrastive learning. Extensive experiments conducted on four HMTC datasets demonstrate the effectiveness of the proposed approach.

Strength: This paper presents a promising method to improve the accuracy of HMTC models. The paper is well-written and easy to follow.  Extensive experiments on four multi-path HMTC datasets are conducted.  All reviewers agree that this is a solid contribution but the excitement is relatively limited.